# Sigma 1 Receptor Is Overexpressed in Hepatocellular Adenoma: Involvement of ERα and HNF1α

**DOI:** 10.3390/cancers12082213

**Published:** 2020-08-07

**Authors:** Laure Villemain, Sylvie Prigent, Aurélie Abou-Lovergne, Laura Pelletier, Magali Chiral, Marco Pontoglio, Fabienne Foufelle, Stefano Caruso, Raphael Pineau, Sandra Rebouissou, Eric Chevet, Jessica Zucman-Rossi, Laurent Combettes

**Affiliations:** 1UMRs 1174, University Paris-Saclay, Inserm, 91405 Orsay, France; villemain_laure@yahoo.fr (L.V.); sylvie.j.prigent@wanadoo.fr (S.P.); aurelie.aboulovergne@gmail.com (A.A.-L.); 2Centre de Recherche des Cordeliers, Sorbonne Université, Inserm, Université de Paris, Université Sorbonne Paris Nord, Functional Genomics of Solid Tumors laboratory, F-75006 Paris, France; laura.pelletier@inserm.fr (L.P.); stefano.caruso@inserm.fr (S.C.); sandra.rebouissou@inserm.fr (S.R.); jessica.zucman-rossi@inserm.fr (J.Z.-R.); 3Institut Necker - Enfants Malades (INEM), 14, rue Maria Héléna Vieira da Silva Bâtiment Leriche, 75014 Paris, France; magali.chiral@inserm.fr (M.C.); marco.pontoglio@inserm.fr (M.P.); 4Centre de Recherche des Cordeliers, Sorbonne University, Inserm, U1138” Metabolic diseases, diabetes and co-morbidities” F-75006 Paris, France; fabienne.foufelle@crc.jussieu.fr; 5UMRs 1242 “Chemistry, Oncogenesis, stress, Signaling” (COSS), University de Rennes-1, 35042 Rennes, France; raphael.pineau.1@univ-rennes1.fr (R.P.); eric.chevet@inserm.fr (E.C.); 6Centre de Lutte Contre le Cancer Eugène Marquis, 35042 Rennes, France; 7Hôpital Européen Georges Pompidou, AP-HP, F-75015 Paris, France

**Keywords:** hepatocyte, liver tumor, steatosis, proliferation

## Abstract

Sigma receptor 1 (SigR1) is an endoplasmic reticulum resident integral membrane protein whose functions remain unclear. Although the liver shows the highest expression of SigR1, its role in this organ is unknown. SigR1 is overexpressed in many cancers and its expression is correlated to hormonal status in hormone-dependent cancers. To better understand the role of SigR1 in hepatocytes we focused our work on the regulation of its expression in tumoral liver. In this context, hepatocellular adenomas, benign hepatic tumors associated with estrogen intake are of particular interest. The expression of SigR1 mRNA was assessed in hepatocellular adenoma (HCA) patients using qPCR. The impact of estrogen on the expression of SigR1 was studied in vivo (mice) and in vitro (HepG2 and Huh7 cells). The effect of HNF1α on the expression of SigR1 was studied in vivo by comparing wild type mice to HNF1 knockout mice. Estrogen enhanced SigR1 expression through its nuclear receptor ERα. HNF1α mutated HCA (H-HCA) significantly overexpressed SigR1 compared to all other HCA subtypes. HNF1 knockout mice showed an increase in SigR1 expression. Overexpressing SigR1 in cellular models increases proliferation rate and storage of lipid droplets, which phenocopies the H-HCA phenotype. SigR1 is involved in hepatocyte proliferation and steatosis and may play an important role in the control of the H-HCA phenotype.

## 1. Introduction

SigR1 is a transmembrane protein of the endoplasmic reticulum (ER), located in the mitochondria-associated membranes (MAM) and has been also described in the plasma membrane of Xenopus oocytes [1]. SigR1 is ubiquitous but its highest expression is in the central nervous system (CNS) and the liver. There is no endogenous ligand identified today, but neurosteroids such as progesterone and pregnenolone have been found as potential natural ligands of SigR1 [2].

Regarding its role in the cell, SigR1 has been described as a Ca^2+^ sensitive chaperone protein ensuring an important role in the MAM where it appears to play an important role as gatekeeper to keep ER stress under control [3,4]. Under physiological conditions, SigR1 is associated with BiP, also known as GRP78, another ER-resident chaperone protein [3,4]. Once activated, it can dissociate from BiP and chaperone client proteins [4,5] such as the InsP3 receptor [6]. More recently, it has been suggested that SigR1 would represent a new type of protein, which was named “pluripotent modulators” [7]. This is based on the fact that it can bind and modulate different classes of protein in various locations of the cell. Moreover, depending on the cell type, its actions and partners vary [7].

SigR1 has mostly been studied in the CNS and has been linked to several neurodegenerative diseases such as Alzheimer’s, Parkinson, or Huntington diseases [8]. Mutation of the SigR1 gene was also found to cause a juvenile form of amyotrophic lateral sclerosis [9]. Regarding peripheral tissues, binding experiments conducted in the 1990s revealed a very high expression of Sigma receptors in many tumor cell lines from various human cancer tissues [1,10]. A specific SigR1 antibody showed a high level of SigR1 in lung, breast, and prostate cancer cell lines compared to the basal level [11]. However, the role of SigR1 in cancer is not clear as some studies showed that silencing SigR1 by siRNA in human breast cancer cell line significantly reduced cell proliferation [11], while other reports suggest that SigR1 could inhibit tumor growth [12,13]. Likewise, it has been reported that the level of expression of SigR1 was related to the aggressiveness of cancer: the more SigR1 is expressed, the more aggressive the tumor is [11,14]. In contrast, Simony-Lafontaine et al. established that in human breast tumors, the level of SigR1 was correlated to patient survival, suggesting that SigR1 inhibits tumor growth [12] and more recently, a decrease in the expression of SigR1 in Hepatocellular carcinoma [15] has been shown. Nevertheless, many studies show the involvement of SigR1 in cancer [16,17], and, for example, SigR1 has been suggested as a potential prognosis and diagnostic marker for breast tumors [12] and prostate cancers [18]. Interestingly, in breast and prostate tumors, the SigR1expression is respectively related to progesterone and androgen status [11,12] suggesting a potential role of these hormones on the expression of SigR1.

Although SigR1 is highly expressed in the liver, its role in this organ is yet to be determined. As SigR1 is overexpressed in many tumors, we focused our investigation on the expression of SigR1 in tumoral livers. As there might be a role of hormones for the expression of SigR1, Hepatocellular adenomas (HCAs) are of particular interest because they correlate with estrogen exposure. Indeed, it is well known that prolonged use of oral contraceptive steroids is a common risk factor for the development of HCAs [19]. As a consequence, we first evaluated the pathological context in which SigR1 is overexpressed and identified the cause and functional consequences of this overexpression in model systems and cultured cells, respectively.

## 2. Results

### 2.1. HCA Overexpress SigR1

HCAs are benign liver tumors resulting from the proliferation of hepatocytes and presenting moderate to severe steatosis [19,20,21]. The sample series were obtained from previously published data [19]. Some of these samples were used to measure SigR1 using qPCR and listed in Appendix A. Analysis of the results indicated that SigR1 was significantly upregulated in HCAs compared to the non-tumoral liver (Figure 1). Provided that the etiology of HCA has been shown to relate to estrogen exposure [22], we then asked whether estrogen could impact on the expression of SigR1 in hepatocytes.

### 2.2. Estrogens Induce Expression of SigR1

The effect of estrogen on the expression of SigR1 was evaluated on male mice to avoid the variation of estrogen levels which occurs during the female mice estrous cycle [23]. Male mice were injected with a 10 mg/kg solution of 17βestradiol (E2) and livers were removed 24 h later. The expression of SigR1 was quantified then as described in the methods section. As shown in Figure 2A, the expression of SigR1 increased significantly by 1.8-fold after E2 treatment. Enhanced expression of SigR1 was also observed in Huh7 cells exposed to E2 for 24, 48, and 72 h (Figure 2B). Taken together, these results show that estrogens are activators of SigR1 expression in hepatocytes.

### 2.3. Estrogens Use ERα to Regulate the Expression of SigR1

We then investigated the mechanism of action of estrogens on SigR1 expression. Estrogens exert their functions through three different receptors: a G protein-coupled receptor, GPR30, and two nuclear receptors, ERα and ERβ, the latter not being expressed by hepatocyte [24]. As ERα is a ligand-activated transcription factor, we thus searched for putative ERα binding sites on the SigR1 promoter using bioinformatic analysis. This showed several putative binding sites for ERα but only one of them showing a highly probable ERα binding site (ERE: ER responsive element, Appendix A). To determine if this site was involved in the control of SigR1 expression, we used a reporter gene with luciferase under the control of the SigR1 promoter. Huh7 cells were thus transfected with either the entire SigR1 promoter or the promoter truncated, missing the ERE, and luciferase activity was quantified (Figure 3A). Luciferase activity was four times lower in cells expressing the truncated promoter than in those transfected with the whole promoter, suggesting that SigR1 promoter activity depends in part on estrogens. To confirm these results, we constructed a plasmid containing the SigR1 promoter mutated within the ERE (Figure 3B). The results indicated that when ERE was mutated, the SigR1 promoter activity decreased by threefold. We subsequently used HepG2 cells which do not express ERα, to confirm our results obtained in Huh7 cells. HepG2 were transfected or not with ERα together with the reporter gene luciferase under the control of the SigR1 promoter, which was mutated, or not, for ERE and luciferase activity was quantified (Figure 3C). Cells were also treated with E2. ERα transfection increased SigR1 expression in HepG2 cells suggesting that ERα activates SigR1 promoter activity. This increase was even higher when the transfected HepG2 cells were treated with E2 (10 nM).

As expected, the mutation of the putative binding site for ERα on the SigR1 promoter abolished E2 effects on SigR1 expression, confirming that estrogens may act on SigR1 promoter through ERα. Inhibition was not complete, suggesting that there may even be at least one other ERα binding site on the Sig1 promoter. The binding of ERα on the ERE of the SigR1 promoter was confirmed using chromatin immunoprecipitation (ChIP) experiments in HepG2 cells transfected with SigR1 wild-type or mutated promoter and ERα (Figure 3D). As shown in Figure 3D, ERα can bind SigR1 promoter, but, when ERE is mutated, the binding is decreased, confirming the presence of a functional ERE on the SigR1 promoter. Taken together, our results demonstrate that estrogens induce the expression of SigR1 in hepatocytes via ERα.

### 2.4. H-HCA Overexpress SigR1 Compared to Other Subgroups of HCA

In recent years, the molecular analysis of HCAs observed in patients allowed to classify HCAs according to different molecular categories [19]. Among these different subtypes, HCAs with inactivating mutations of hepatocyte nuclear factor 1a (HNF1α; H-HCA) are the most numerous and account for 40% of all HCAs. We thus analyzed the expression of SigR1 mRNA in these HCA subtypes and found that the H-HCAs, which are caused by a bi-allelic inactivating mutation of HNF1α, showed significantly higher levels of SigR1 mRNA compared to non-tumoral liver and all other HCA subtypes (Figure 4A and Appendix A for a detailed analysis). Western Blot analysis of liver samples from patients with HNF1α mutation confirmed that HNF1α loss of function correlated with increased SigR1 expression (Figure 4B).

HNF1α is a transcription factor involved in several metabolic pathways in hepatocytes [20] and is mutated in type 2 diabetes MODY 3 (Maturity Onset Diabetes of the Young 3) [25]. This type of HCA is mostly found in women under oral contraceptive treatment [19]. These observations prompted us to test whether the loss of HNF1α has an effect on SigR1expression in H-HCAs.

### 2.5. Loss of HNF1α Leads to Increased SigR1 Expression

In patients, SigR1 overexpression seems to be correlated to the loss of HNF1α in hepatocellular adenomas. To confirm this hypothesis, we assessed SigR1 expression in mice lacking HNF1α [26]. These mice suffered from type 2 diabetes and presented hepatomegaly and liver steatosis [26,27]. Quantitative RT-PCR and western blot experiments on mouse liver samples showed a higher SigR1 level in HNF1α KO mice compared to the control mice (Figure 5A,B). These results confirmed that the absence of functional HNF1α correlated with an increase in SigR1 expression, suggesting an inhibitory effect of HNF1α on SigR1. HNF1α can act as a transcriptional repressor [27], and bioinformatic analysis shows a putative binding site for HNF1α on the SigR1 promoter (Supporting Figure 1). HNF1α is also involved in various metabolic pathways potentially impacting SigR1 expression [26,28]. Of note, it has been shown that most HNF1α target genes in hepatocytes are only slightly affected in KO-HNF1α mice [26]. This may reflect the fact that HNF1β, present at a low level in hepatocytes, could assume the function of the alpha protein [29]. This could also be explained by a more complex indirect interaction between SigR1 and HNF1α, as it has been shown on other promoters [30].

In summary, estrogens exert a positive effect on SigR1 expression whereas HNF1α exerts a negative effect. These two situations are found in patients with H-HCA and probably explain why it is in this population that we observe the strongest overexpression of SigR1. We thus investigated the consequences of overexpression of SigR1 in hepatocytes.

### 2.6. SigR1 Overexpression Affects Cell Proliferation

According to the literature, SigR1 is linked to cancer and cell proliferation [1]. Furthermore, HNF1α KO mice, which overexpress SigR1, have important hepatomegaly due to hepatocyte proliferation [19] and HCAs derive from a benign proliferation of hepatocytes [19]. These observations led us to assess the effect of overexpression of SigR1 on hepatocyte proliferation. We thus established stable clones of HepG2 cells overexpressing SigR1. Among the numerous stably transfected lines that we obtained, three clones (clone 9, 10, and 11) showed the most appropriate expression of myc-tagged SigR1. We then measured their proliferation rates. The results showed a higher proliferation rate in cells overexpressing SigR1, than in control HepG2 cells (Figure 6A) and the more SigR1 was overexpressed, the higher the proliferation rate was (Figure 6B**).** Because the growth kinetic was faster for clone 9 (see Figure 6B), we, therefore, chose this clone for the rest of the study. Additionally, we observed that decreasing the expression of SigR1 using two different siRNA directed against SigR1 brought these cells back to their basal proliferation rate (Figure 6C). Similar results were observed in Huh7 cells (Appendix A). All these results suggest that SigR1 is involved in hepatocyte proliferation.

### 2.7. SigR1 Overexpression Affects Lipid Storage

Since H-HCA is also characterized by marked steatosis [21,22], we then assessed the effect of SigR1 overexpression on lipid storage by hepatocytes. We first measured the accumulation of lipid droplets in HepG2 cells, overexpressing SigR1 (Clone 9) or not (HepG2) using Oil Red O staining (Figure 7A). Quantification of the area occupied by lipid droplets in these cells revealed that it was significantly increased in Clone 9 compared to HepG2 cells (Figure 7B). However, the number of lipid droplets in these cells was not different when SigR1 was overexpressed (Figure 7C), indicating an increase in droplets size. In Huh7 cells, both accumulation of lipid droplets and a significant increase of the area occupied by the droplets were also observed in cells overexpressing SigR1 (Huh7 SigR1c-Myc) (Figure 7D,E). Interestingly, in Huh7 cells, there was also a significant increase in the number of droplets when SigR1 was overexpressed (Figure 7F). Taken together, these results suggest that, overexpression of SigR1 induces an increase of lipid storage by the hepatocytes. These results combined with the fact that HNF1α KO mice suffer from liver steatosis [26,27] led us to correlate steatosis in HCA patients with SigR1 expression (Figure 7G). HCA were sorted into four groups according to their level of steatosis: no steatosis, less than 1/3, between 1/3 and 2/3, and over 2/3 of steatosis (see Appendix A). The results showed that the level of SigR1 mRNA was positively correlated with the degree of steatosis (Figure 7G). All these results strongly suggest a role of SigR1 in hepatic steatosis.

## 3. Discussion

In this work, we have investigated the regulation and function of SigR1 in the liver and in particular in hepatocellular adenoma.

In a study carried out on liver tissue from 30 patients (26 men and 4 women) with hepatocellular carcinoma (HCC), a decrease in the expression of SigR1 was observed compared to the non-tumoral liver [15]. This result contrasts with what we observed in HCA. However, HCA and HCC have been characterized as very different types of tumors, the former being benign (in a majority of cases) and the latter being aggressive and leading to the patient’s death in most cases. As such, HCA and HCC cannot be directly compared. Moreover, we showed that in HCA the significant increase of SigR1 expression was linked to the inactivation of HNF1α (see Figure 4A), which is mostly associated with the consumption of oral contraceptives and found consequently mostly in women. In the manuscript by Xu and colleagues, beyond the fact that HNF1α status was likely wild-type (as in most HCC), 29 HCC were analyzed, including 26 from men while our study focused on 349 HCA, 85% of which developed in women. Furthermore, the etiology of HCC was not well defined but could be linked to cirrhosis [15], which is not an underlying condition state in HCA. Finally, the opposite effect of overexpression of SigR1 on the proliferation of HepG2 cells could be because the cells may originate from different sources and exhibit different characteristics [31]. Note that overexpression of SigR1 in HuH7 also induced an increase in the proliferation rate of these cells (Appendix A), which is a confirmation in another cell line with a different genetic background (p53 mutation in HuH7 cells vs. beta-catenin truncation in HepG2).

### 3.1. SigR1 Expression is Upregulated by Estrogens in Hepatocytes

SigR1 is highly expressed in hepatocytes but its role in the liver remains poorly explored. Our study demonstrated that SigR1 was significantly overexpressed in certain types of HCAs, especially in H-HCA, benign tumor characterized by steatosis [19,20,21].

HCAs are mostly found in women under oral contraceptive treatments [19]. SigR1 overexpression was also observed in breast and prostate cancers, both hormone-dependent tumors [12,18]. It has been shown that steroid hormones could directly bind SigR1 [2] and progesterone has even been suggested as an endogenous ligand of SigR1 [2,5]. Herein, we showed that estrogens can induce SigR1 expression via the ERα pathway. This effect depends on the nuclear receptor of estrogens and not the GPR30 membrane receptor since the addition of E2 to HepG2 cells, which show endogenous expression of GPR30, but not ERα [32], does not affect SigR1 expression.

### 3.2. SigR1 Expression is Repressed by HNF1

The major feature of HCAs that overexpress most SigR1 is an inactivation through biallelic mutations of the *HNF1α* gene [19]. We also showed that HNF1α knock out leads to SigR1 overexpression in mouse liver. This could occur through a direct mechanism, as there is a putative binding site for HNF1α on the promoter of SigR1, suggesting that HNF1α could be a repressor of SigR1. An inhibitory effect of HNF1α has for instance already been shown on the transcription factor and lipid sensor PPARγ (peroxisome proliferator-activated receptor γ) [27]. Another argument in favor of this hypothesis is that HNF1α can interact with an amino-terminal enhancer of split (AES) which is a co-repressor and repress HNF1α-mediated transcription [33]. Yet the action of HNF1α on the expression of SigR1 could also be indirect. HNF1α regulation is complex, involving in particular HNF4α, another transcription factor which expression precedes HNF1α in embryos and can bind HNF1α promoter and induces its transcription [29]. In return, HNF1α inhibits HNF4α expression [29]. It has been shown that most of the HNF1α target genes can also bind HNF4α [34]. This complex network of transcription factors in hepatocyte could explain why we could not observe any effect of siRNA against HNF1α in cellular models (not shown).

Moreover, an interplay between HNF1α and estrogens and SigR1 must be also suggested. It has been shown that HNF1α loss of function leads to a decrease of CYP1A1, CYP1A2, and CYP3A4. These cytochromes are responsible for the oxidation of estradiol in the hepatocyte [20], therefore promoting the passage of estradiol in an inactive form. These observations were confirmed in cultured hepatocyte in which HNF1α was inhibited [20]. These results could explain the observed activation of SigR1 by estrogens in H-HCA. Indeed, the loss of function of HNF1-alpha, by decreasing the detoxification of estradiol, would induce an increase of the latter in its active form. Thus H-HCA patients would be more sensitive to estrogens [20]. This higher sensitivity could account, at least in part, for overexpression of SigR1 observed in H-HCA.

### 3.3. SigR1 Overexpression Controls Hepatocyte Proliferation and Fat Accumulation

Next, by investigating the consequences of SigR1 overexpression in hepatocytes, we showed that this induced cell proliferation in Huh7 and HepG2 cells. These experimental results suggest that SigR1 overexpression may contribute to benign liver tumorigenesis related to the loss of HNF1A function, as well as, the proliferation of hepatocytes in HNF1α-KO mice and thereby reinforce the idea of a role of SigR1 in proliferation already discussed [1,35]. SigR1 is described as a protein of MAMs, yet it now appears that MAMs play a vital role in the regulation of fundamental physiological processes such as calcium signaling and lipid homeostasis [36], two pathways controlling cell fate. Moreover, it was shown that, in tumor cells, the MAM landscape was modified inducing a change in Ca^2+^ flow between ER and mitochondria [36]. The effect of the overexpression of SigR1 on cell proliferation could result from a modification of calcium signal at MAM, as SigR1 can modulate the conductivity of the InsP3-R3, facilitating Ca^2+^ transfer from the ER to mitochondria [6]. In breast cancer, SigR1 in the MAM is upregulated [10] and favors cell migration by regulating Ca^2+^ homeostasis in association with the SK3 channel (calcium-activated K+ channel) and Orai1 [14]. The effect of SigR1 on cell proliferation could also result from its interaction with IRE1 [37], which has been linked to cell proliferation in several cancers including colon cancer, breast cancer, prostate cancer, and melanoma [38].

Another major process regulated in the MAM is lipid synthesis and transportation [5,39]. This is of particular interest as H-HCAs and HNF1α-KO mouse livers are characterized by steatosis [19,26]. Furthermore, we showed that overexpression of SigR1 induced the storage of lipid droplets by HepG2 and Huh7 cells. These results suggest SigR1 plays a key role in steatosis. As a MAM protein, SigR1 has already been associated with lipid metabolism [40,41]. SigR1 has been associated with cholesterol synthesis as silencing SigR1 induced a decrease in pregnenolone synthesis [42]. Moreover, it has been demonstrated that ER stress induces a BiP/SigR1 dissociation to let SigR1 act like a chaperone [6]. ER stress is known to be involved in steatosis; it promotes de novo lipogenesis in hepatocytes [43]. In hepatocytes, SigR1 could for example chaperone a protein involved in de novo lipogenesis, such as INSIG1 (insulin-induced gene 1 protein). This resident RE protein may be involved in this de novo lipogenesis [44] and it has been suggested that SigR1 interacts with INSIG1 [45]. Furthermore, because SigR1 has been described as a “hub protein”, or a “pluripotent modulators”, it would be interesting to study the landscape and the expression of proteins of MAMs in HCAs as well as systematically characterizing the SigR1 interactome in this disease.

## 4. Materials and Methods

### 4.1. Human Tissue Samples and Animals

A total of 391 fresh-frozen tissue samples, including 349 HCA, were included in this study (Appendix A). Patients and tumor features were described previously [19]. All animal procedures met the European Community Directive guidelines (Agreement B33-522-2/ No DIR 1322) and were approved by the local ethics committee. All patients gave informed consent according to French law and Paris Saint-Louis Institutional Review Board committee approved this study (Paris Saint-Louis, 2004; INSERM IRB 2010; the French Liver Biobanks Network, AFAQ NF S96-900; and Hepatobio Bank). Three independent sets of experiments were carried out on 8-week-old C56/Bl7 male mice housed in ventilated racks. Mice were injected with a 10mg/kg solution of 17βestradiol (E2) in sunflower oil as a vehicle in IP. Twenty-four hours after injection mice were sacrificed and the liver was removed to extract RNA and proteins. HNF1α KO mice line has been previously described [20], mice were 4 to 5 months old when sacrificed.

### 4.2. Cell Culture and Proliferation Assay

HepG2 and Huh7 cell lines were maintained in Dulbecco’s Modified of Eagle’s Medium (DMEM, ThermoFisher, (Illkirch, Cedex France) containing 10% fetal bovine serum and 1% antibiotics (Penicillin, Streptomycin, and Fungizone) in 5% CO_2_ at 37 °C. To measure proliferation, HepG2 and Huh7 cells were plated in a 6-well plate at 3.10^5^ cells/well. Cells were then counted every day for 5 days using a hemocytometer.

### 4.3. Transfection with Luciferase Reporter and ERα Overexpression

Constructs containing the Firefly Luciferase under the control of the SigR1 wild-type or mutant promoter in pGL4.10 were synthesized in the lab. Truncated mutants were made with PCR and then cloned into pGL4.10. The punctual mutation was made with QuickChange Lightning Site-Directed Mutagenesis (Agilent Technologies, Les Ulis, France). These constructs were transfected in both Huh7 and HepG2 cells using Fugene HD (Promega, Madison, WI, USA) according to the supplier’s recommendations. In brief, 2.5 µg of each plasmid was transfected with 3 µL of Fugene and 25 ng of pRL-TK containing Renilla Luciferase (internal control). Luciferase activity was determined by luminometry (Wallac 1420, Perkin Elmer, Villebon-sur-Yvette, France) 48 h after transfection using the Dual-Luciferase Reporter Assay System Kit (Promega). Relative firefly luciferase activity was calculated upon Renilla activity.

### 4.4. 17βestradiol Treatment

Huh7 and HepG2 cells were plated in six-well plates (4.10^5^ cells/well) in complete DMEM. 24 h later the medium was changed for DMEM without Phenol Red (Thermofisher) containing 10% depleted steroid serum (EUROBIO, Les Ulis, France). HepG2 cells were transfected with pcDNA3ERα as they do not express ERα. Huh7 and transfected HepG2 cells were treated with 10 nM 17βestradiol in ethanol as a vehicle; the choice of these E2 concentrations has been dictated by the literature [46]. After 24, 48, or 72 h of treatment, cells were lysed to assess SigR1 expression using Western Blot.

### 4.5. Generating Myc-SigR1 Stably Expressing HepG2 Cell Line

The human SigR1 (U75283) subcloned into the pcDNA3 vector was obtained from Dr. F Monnet. SigR1 cDNA was digested by HindIII and XbaI and ligated into the pCMV-cMyc-C vector (Stratagene, Santa Clara, CA, USA) to express an N-terminal cMyc-tagged SigR1 protein. Plasmid transfection was carried out either with the empty vector (controls) or with the same vector carrying the S1R cDNA using calcium phosphate in HepG2 cells. The clonal selection was performed using 600 µg/mL G418. SigR1 expression was monitored using immunofluorescence with anti-cmyc antibodies.

### 4.6. Protein Extraction and Western Blotting

Whole-cell lysates were obtained with a boiling lysis buffer (1% SDS, 1mM Tris HCL, and 8mM Sodium orthovanadate). Liver samples were washed with PBS 1X and then grinded in lysis buffer (10 mM HEPES, 0.2 mM EDTA, 10 mM KCL, 1.5 mM MgCl_2_, 150 mM NaCl, 0.01 mM DTT, 0.2% NP40 and 40U RNase out). Lysates were centrifuged at 10,000 g for 5 min at 4 °C. 40 µg of protein extracts were resolved on a 12% SDS-page gel and transferred on a nitrocellulose membrane. Membranes were blocked in 5% skim milk (2.5% for SigR1 antibody), then incubated overnight at 4 °C with specific primary antibodies: SigR1 (1/400; Covalab), ERα (1/400; H184 Santa Cruz), HNF1α (1/500; Santa Cruz). The bound antibodies were detected (Fusion, Vilbert-Lourmat, France) using corresponding horseradish peroxidase-conjugated secondary antibodies and chemiluminescence detection kit (Biorad, Marnes-la-Coquette, France). Quantification of the signals was performed by densitometry using the ImageJ software.

### 4.7. RT-qPCR

For human tumors analyses, *mRNA* expression levels were assessed by quantitative RT-PCR using Fluidigm 96.96 Dynamic Arrays and specific TaqMan predesigned assays (SIGMAR1 = Hs00195337_m1) (Life Technologies, Carlsbad, CA, USA). Data were calibrated with the RNA ribosomal 18S and changes in mRNA expression levels were determined using a comparative CT method using 5 normal tissue samples as control [19]. For mouse liver tissues and cell lines, total RNA was extracted using TRI Reagent (Sigma, Lyon, France), according to the manufacturer’s protocol. cDNA was amplified using PCR in the presence of independent forward and reverse primers detailed in Appendix A. For quantitative PCR, cDNA was amplified using the SYBR green PCR kit (Bio-Rad) and normalized to cyclophilin using Opticon Monitor 3 software (Bio-Rad). PCR conditions were as follows: 95 °C 10 min, then 40 cycles at 95 °C for 30 sec and at 60 °C for 1 min.

### 4.8. Oil Red O Staining

HepG2 et Huh7 were plated at 10^5^ cells per well in a six-well plate. After 3 to 6 days, cells were washed 3 times with cold PBS, fixed in 10% formaldehyde, and then washed twice with 60% isopropanol. Oil Red O solution is added for 15 min at room temperature before being washed twice with cold PBS.

### 4.9. Chromatin Immunoprecipitation (ChIP)

HepG2 cells were seeded in 6 well plates and transfected with pcDNA3ERα as previously described to express ERα. ChIP were done with Pierce Agarose ChIP Kit (ThermoFisher) and primary antibody against ERα H184 (Santa Cruz Biotechnologies, Santa Cruz, CA, USA). Purified DNA was amplified using PCR in the presence of independent forward and reverse primers for the SigR1 promoter detailed in Appendix A. Total input was determined using an antibody against PolII that binds GAPDH promoter, antibody, and primers provided with the kit.

### 4.10. Statistical Analyses

For human samples: Data visualization and statistical analysis were performed using R software version 3.5.1 (R Foundation for Statistical Computing, Vienna, Austria. “https://www.R-project.org)” and Bioconductor packages. Comparisons of the mRNA expression levels between groups were assessed using the Mann–Whitney U test (2 groups) or Kruskal–Wallis Test (more than two groups). Correlation analysis between continuous variables was performed using the Pearson r correlation when both variables were normally distributed with the assumptions of linearity and homoscedasticity. *P*-value < 0.05 was considered as significant. For mouse tissues and cell lines, statistical analysis was performed using Prism software, a comparison between two groups was assessed using t-Test. *P*-value < 0.05 was considered as significant. * *p* < 0.05; ** *p* < 0.01; *** *p* < 0.001.

## 5. Conclusions

To conclude, an intake of estrogens associated with a loss of function of HNF1α in H-HCA leads to SigR1 overexpression. In turn, this overexpression may lead to an increase in hepatocyte proliferation and steatosis, as observed in hepatoma cell lines. These results are consistent with the hepatocyte proliferation and steatosis observed in patients with H-HCA. SigR1 would, therefore, be a potential target in the treatment of cancers but also for fatty liver. Hepatic steatosis is an increasingly widespread disease, in 2017 for example, an estimated 900,000 patients were affected in France and 6 million in the United States. Since many SigR1 ligands are already developed by the pharmaceutical industry to treat psychiatric disorders, it would be interesting to test the effects of theses ligands on steatosis and proliferation in the context of cancerous tumors.

## Figures and Tables

**Figure 1 cancers-12-02213-f001:**
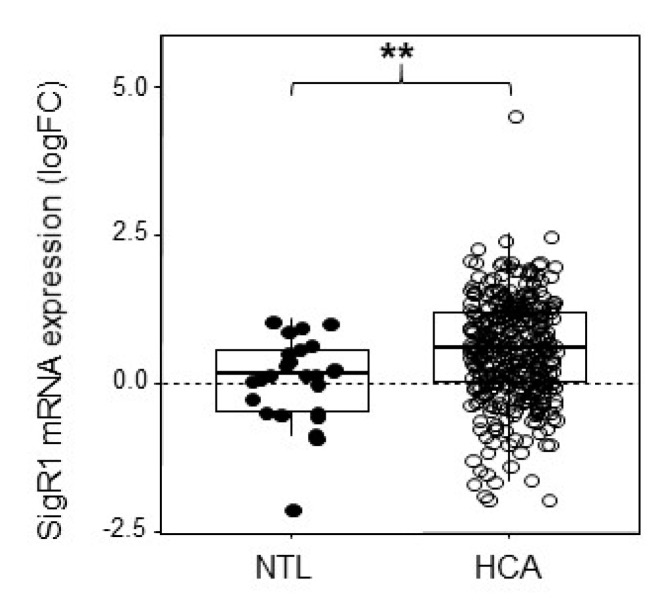
Level of expression of SigR1 in hepatocellular adenoma (HCA). Box-and-whisker plot of mRNA expression of SigR1 in non-tumoral liver (NTL, solid dots, *n* = 35) and HCA (empty dots, *n* = 349). *P* = 8.4 × 10^−9^. Data are shown as mean ± SEM, Mann–Whitney U test. ** *p* < 0.01.

**Figure 2 cancers-12-02213-f002:**
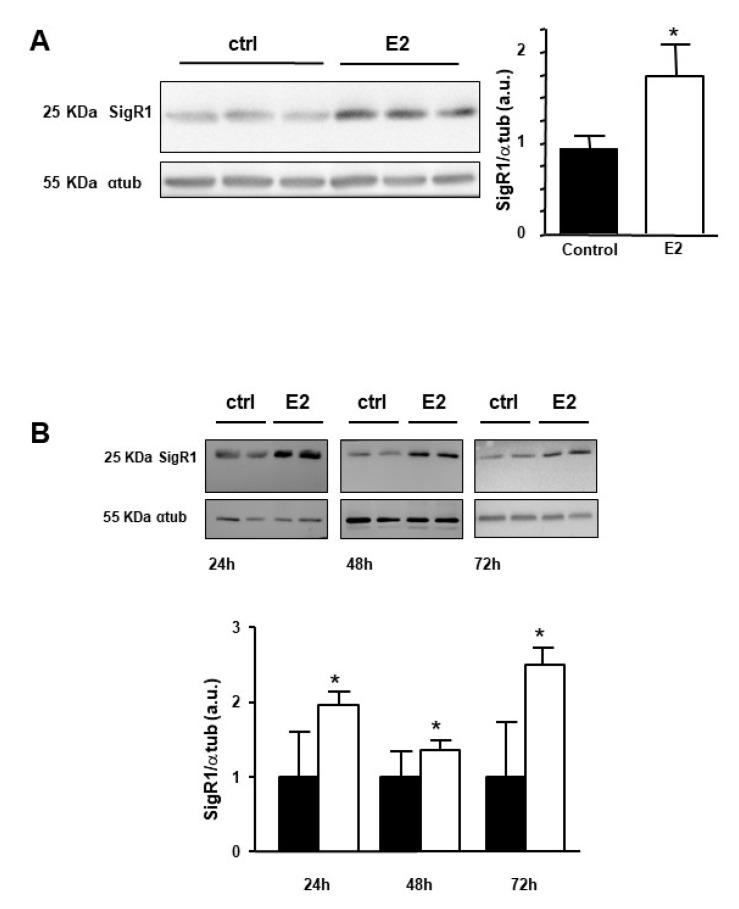
Effect of estrogens on the expression of SigR1. (**A**) Protein expression of SigR1 by western blot and quantification in control mice liver samples (ctrl, solid bars, *n* = 19) or 24 h after the injection of E2 (E2, empty bars, *n* = 20), *p* = 0.03. (**B**) Protein expression of SigR1 by western blot quantification on Huh7 cell lysate in control (ctrl, solid bars, *n* = 10) and after 24, 48, or 72 h treatment with E2 (empty bars, *n* = 5). Data are shown as mean ± SEM, 24 h—*p* = 0.001, 48 h—*p* = 0.005, 72 h—*p* = 0.002. Mann–Whitney U test. * *p* < 0.05, More details of the western bolts, please view at the Appendix A.

**Figure 3 cancers-12-02213-f003:**
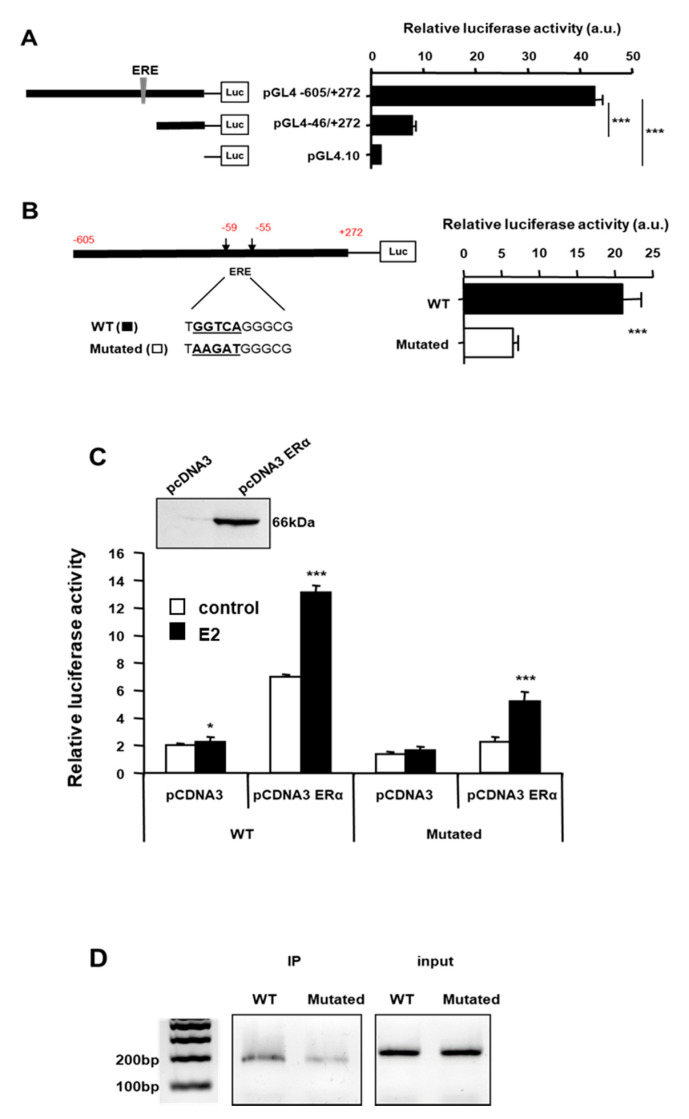
The activity of the SigR1 promoter. Relative luciferase activity was quantified in Huh7 cells after transfection with a reporter gene luciferase under the control of the SigR1 promoter, with or without the part containing the putative ERE (**A**) and with or without a mutated ERE (**B**). (**C**) HepG2 cells were transfected or not with ER1α (+/− pcDNA3 ERα) and then stimulated (solid bars) or not (empty bars) with E2. Luciferase activity was quantified with the Wild Type (WT) or mutated (Mutated) SigR1 promoter. (**D**) Chromatine Immuno Precipitation on HepG2 cells transfected Wild Type (WT) or mutated (Mutated) SigR1 promoter. (**A**–**C**) Data are shown as mean ± SEM, * *p* < 0.05; *** *p* < 0.001, *n* = 6, Mann–Whitney U test.

**Figure 4 cancers-12-02213-f004:**
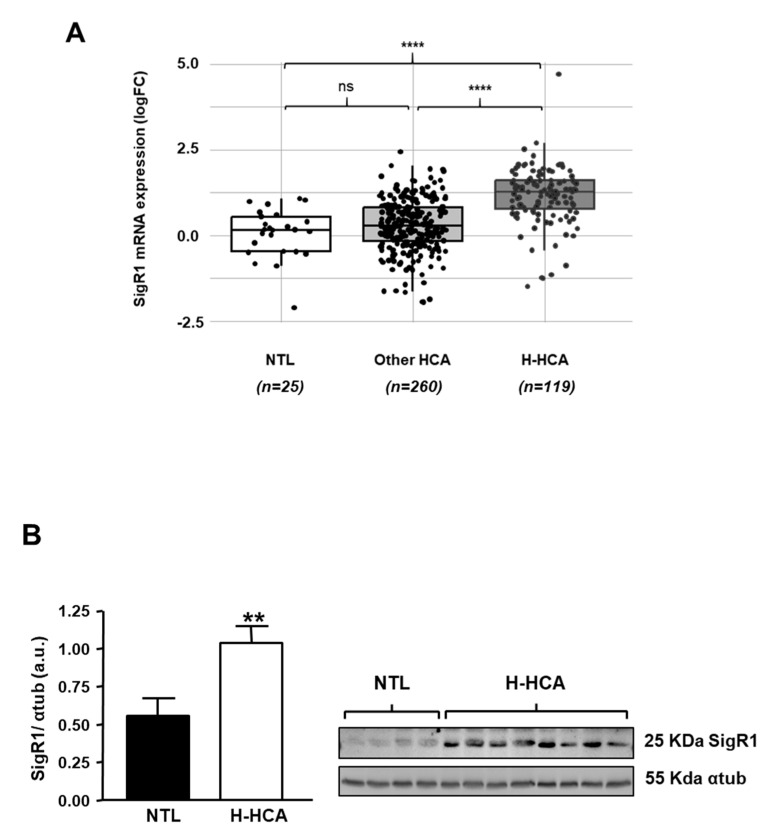
Expression of SigR1 in HCA subtypes. (**A**) Relative mRNA expression of SigR1 in H-HCA subtype (*n* = 119) compared to non-tumoral liver (NTL, *n* = 25) and other HCA (*n* = 260) (see Appendix A). *p* = 7.20 × 10^−9^ (Wilcoxon rank sum test). (**B**) Protein expression of SigR1 by Western Blot and quantification in non-tumoral liver (NTL, *n* = 7) compared to H-HCA (*n* = 12). *p* = 0.0069. Mann–Whitney U test. ** *p* < 0.01; **** *p* < 0.0001. More details of western blots, please view at the Appendix A.

**Figure 5 cancers-12-02213-f005:**
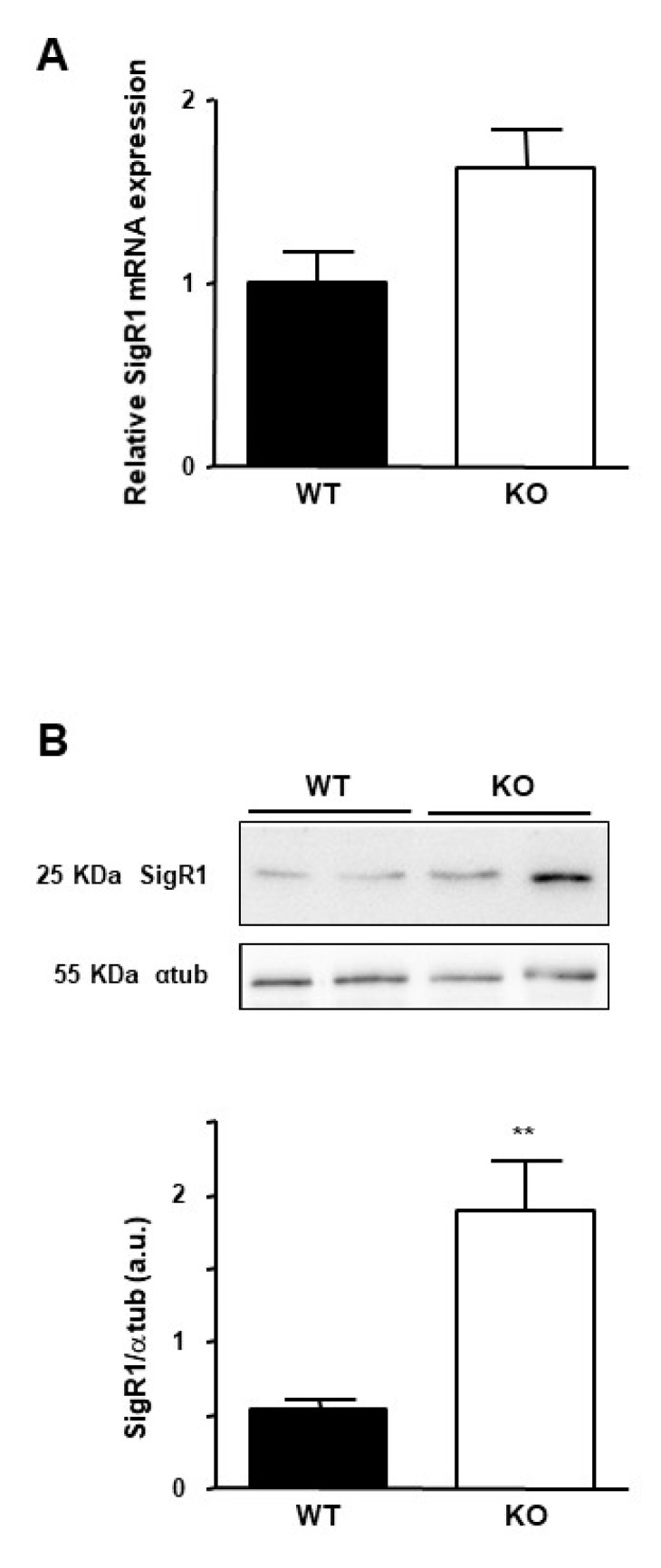
Effect of HNF1α mutation on the expression of the SigR1. (**A**) mRNA expression of SigR1 by qPCR and (**B**) Protein expression of SigR1 by Western Blot on liver samples from Knock out (KO, *n* = 10) and Wild Type (WT, *n* = 9) HNF1α mice. *p* = 0.0028. Data are shown as mean ± SEM, t-Test. ** *p* < 0.01, More details of western blots, please view at the Appendix A.

**Figure 6 cancers-12-02213-f006:**
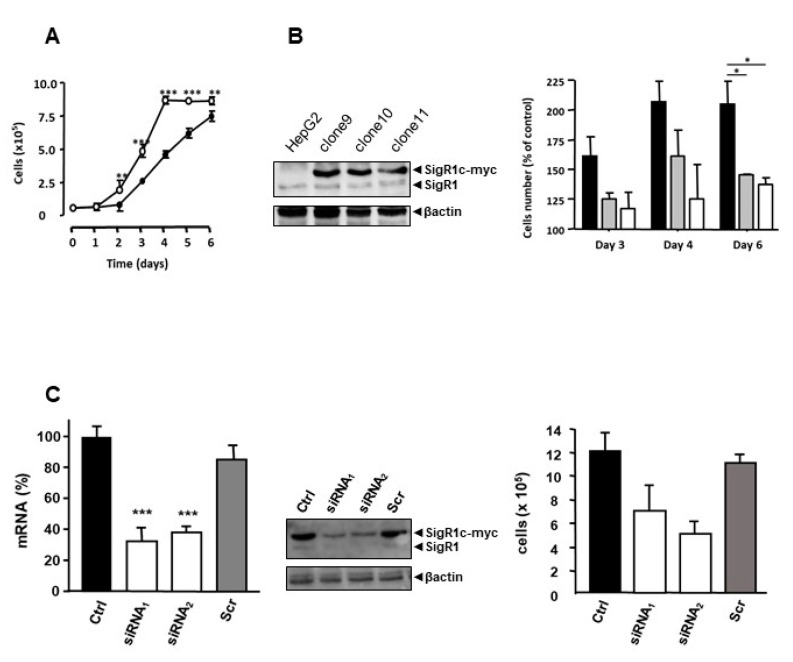
Overexpression of SigR1 increases the proliferation rate of HepG2 cells. (**A**) The proliferation rate of wild type HepG2 cells (black dots) and HepG2 cells that stably overexpress SigR1 (white dots). Results are expressed as means ± SEM (*n* = 6). (**B**) HepG2 cells were transfected with SigR1c-myc to make stable clones (Clone 9, Clone 10, and Clone 11) with differential expression of SigR1. The right panel shows the rate of proliferation of these clones in % of wild type HepG2 cells. Results are expressed as means ± SEM (*n* = 5). (**C**) mRNA expression of SigR1 by qPCR (left panel) and protein expression of SigR1 and SigR1c-myc by western blot (middle) on Clone 9 transfected or not with two different siRNA directed against SigR1 or with a siRNA scramble (Scr). The right panel shows the rate of proliferation of Clone 9 in these conditions. Data are shown as mean ± SEM (*n* = 5), Mann–Whitney U test. * *p* < 0.05; ** *p* < 0.01; *** *p* < 0.001. More details of western blots, please view at the Appendix A.

**Figure 7 cancers-12-02213-f007:**
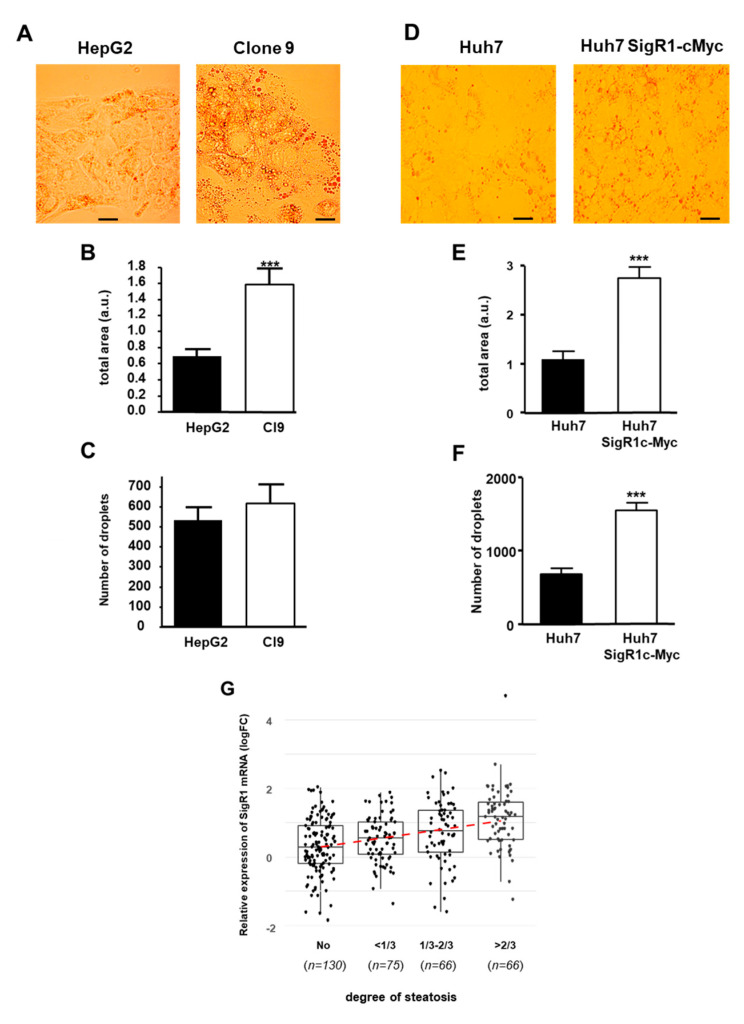
Overexpression of SigR1 increases lipid storage by HepG2 and Huh7 cells. (**A**) Oil Red O staining, (**B**) quantification of staining, *** *p* = 0.0008, and (**C**) count of droplets, *p* > 0.05 of HepG2, and HepG2 overexpressing SigR1 cells (C9). Oil Red O staining (**D**), quantification of staining *p* < 0.0001 (**E**), and count of droplets *** *p* < 0.0001 (**F**) of Huh7 and Huh7 overexpressing SigR1 (Huh7 SigR1c-myc) cells. Data are shown as mean ± SEM, (*n* = 4, > 20 cells for each determination), t-Test. (**G**) The positive correlation between the level of SigR1 mRNA expression and the level of steatosis observed in HCA. HCAs were sorted into four groups according to their level of steatosis (see Appendix A). Linear regression: *p*-value < 0.0001.

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
