# Peer review of "Sigma 1 Receptor is Overexpressed in Hepatocellular Adenoma: Involvement of ERα and HNF1α"

_cancers, 2020, doi:10.3390/cancers12082213_

Round 1

Reviewer 1 Report

This manuscript has been significantly improved.  I agree it for publication.

Reviewer 2 Report

The authors have significantly modified and improved the article since the previous version. This article demonstrates the possible relationship of the SigR1 membrane protein with liver cancer and liver diseases. Being an interesting starting point for further research on their interaction at different stages of these diseases.

My congratulations to all the authors for the work presented.

This manuscript is a resubmission of an earlier submission. The following is a list of the peer review reports and author responses from that submission.

Round 1

Reviewer 1 Report

In this manuscript, L. Villemain et al present their findings about Sigma 1 receptor and hepatocellularadenoma.  The authors described a potential association between SigR1 expression and HCA, and find potential roles ER-alpha and HNF1-alpha in regulating SigR1 expression.  In addition, roles of SigR1 in cell proliferation and fat accumulation were also examined.  The goal of this research is interesting, however, there are still limitations in the current study.

1.  Several key conclusions are opposite to a published research, which is not discussed in the manuscript.  (Xu Q, Li L, Han C, Wei L, Kong L, Lin F. Sigma-1 receptor (σ1R) is downregulated in hepatic malignant tumors and regulates HepG2 cell proliferation, migration and apoptosis. Oncol Rep. 2018;39(3):1405‐1413. doi:10.3892/or.2018.6226)

2.  The purpose of assessing the effects of estrogens is unclear.  Is there any gender disparities in HCA patients and SigR1 expression?  Will HCA male patients show abnormal estrogen levels or ER expression?

3.  Why did the authors choose to give male mice E2 treatment instead of comparing males and females directly?

4.  The HNF1-alpha related study is not relevant to ER related study.  Did the authors observe any correlation between ER signaling and HNF1-alpha activity?

5. Are wild type cells in Figure 6 and 7 transfected with a control clone?

Reviewer 2 Report

The authors present an interesting manuscript in which they begin to define a potential mechanistic role of SigR1 in the development of liver adenomas

Critique:

  1. When the authors analyzed their patient based samples for expression of Sig R1, it appears that they compared non tumoral healthy controls versus HCAs. Did they also compare the levels of SigR1 expression in the adenoma itself versus a disease control ie liver tissue from the same patient that was separate from HCA? Additionally, it appears that much of the Sigr1 elevation occurs in HNF1a mutated HCA patients, should figure 1 be revised to reflect that observation
  2. The in vivo studies in which estrogen injection and SigR1 expression is interesting but is a 24 hour time point following injection – is that reflective of what occurs in patients? What happens in vivo with prolonged exposure to estrogen? As some HCAs arise in females on oral contraceptives, what is the effect of a month long exposure? Similarly, what is the effect of longer term exposure in vitro of the HuH 7 cell cultures? What is the effect in other hepatocyte cell lines beyond HUH 7 cells?
  3. In the knock out HNF1-alpha studies, how do the authors control for diabetes and steatosis and their effects on the hepatic microenvironment in which Sig R1 is thought to have an effect? What was the effect on ER –alpha receptor levels?
  4. For the in vitro work with siRNA, what is the difference between siRNA1 vs. siRNA2? Were different amounts utilized?

Minor editorial issues

Figure 3C - X and Y-axis labels would be helpful

Figure 7G - X axis does not have a label

Reviewer 3 Report

The article entitled Sigma 1 receptor is overexpressed in hepatocellular adenoma: involvement of ERα and HNF1α develops the involvement of the sigma 1 receptor with cancer. The main idea with the conclusions are messy being difficult to follow.

Certain errors are observed such as:

  1. Different type letter in various parts of the text (line 71, 72, 77, 78, 133, 134, 245, 246, 235, 236)
  2. Reference 10 is written twice (line 65)
  3. there is a comma in the reference on line 78
  4. Table 1 should be named as table S1 (line 80)
  5. Missing Y axis in Fig 3.C

Regarding the scientific results, I would like to highlight / ask certain points:

  1. Although significant according to the authors' calculations, a great variety is observed. Could the authors explain why they believe this variety is due? Could it be because of the different subtypes? If so, the idea of ​​an explanation would fit and add an extra graphic in the article.
  2. The authors represent the western blots with several replicas, they should expose a representative blot, since it is confusing.
  3. Fig. 2
  4. What concentration of E2 is injected and added to the cells? Why such concentration? Is there preliminary data?
  5. The statistical study in the case of mice should not be with a t-test, ANOVA should be used.
  6. Fig 4. When dealing with human samples, the statistical study of t-test should not be used.
  7. "This effect could be direct or non-direct" This phrase from the authors should be developed.
  8. Fig 5. ANOVA
  9. Fig. 6
  10. Why the choice of c-myc. Its overexpression alone would lead to an increase in cell proliferation. Have the cars carried out any control only with c-myc?
  11. Some graphics are without legends or graphics making it difficult to follow the text.
  12. Fig. 7 Why did the authors choose clone 9?
  13. The graph of the different subtypes observed in the supplementary data should be included in the main text.

Taking into account the comments, part of the text and the conclusions should be overwritten. Also briefly include an explanation of the importance and possible consequences of said discovery.
